

# Advancing anesthesiology trainee proficiency in airway management *via* simulation-based training: a non-hypoxic apnea duration approach

Lijun Tang[1,*], Yi Zhang[1,*], Lianhua Chen[1], Jinbao Li[1], Shiwei Huang[1], Bin Fang[1], Xiaojing Huang[2] and Lina Huang[1]

[1] Department of Anesthesiology, Shanghai General Hospital, Shanghai Jiao Tong University School of Medicine, Shanghai, China
[2] Department of Pain Medicine, Shanghai Geriatric Medical Center, Shanghai, China
[*] These authors contributed equally to this work.

Corresponding authors
Xiaojing Huang,
anesthesiahuang@163.com
Lina Huang, huanglinash@163.com

## ABSTRACT

**Background**. Effective airway management training for anesthesiology trainees remains crucial for patient safety. While simulation-based education has shown promise, incorporating physiological parameters into training scenarios could enhance learning outcomes.

**Objective**. To evaluate the impact of incorporating non-hypoxic apnea duration awareness in simulation-based airway management education for anesthesiology trainees.

**Methods**. This quasi-experimental study (ChiCTR2200065877) was conducted at Shanghai General Hospital from December 2022 to March 2023. Thirty anesthesiology undergraduates were randomly assigned to either an intervention group, which received non-hypoxic apnea duration information, or a conventional training group. Performance was assessed using a modified Direct Observation of Procedural Skills (DOPS) score and a satisfaction questionnaire. The intervention group was provided with specific non-hypoxic apnea duration data (247 s until $SpO_2$ reached 90%) during simulated difficult airway scenarios.

**Results**. The intervention group demonstrated significantly higher modified DOPS scores ($51.4 \pm 4.4$ *vs.* $43.0 \pm 5.4$, $P < .001$) and satisfaction scores ($45.0 \pm 1.4$ *vs.* $43.1 \pm 2.0$, $P = .005$). Notable improvements were observed in pre-anesthesia preparation ($P = 0.028$), difficult airway management ($P < 0.001$), and crisis response ($P < 0.001$). These findings suggest that incorporating non-hypoxic apnea duration awareness enhances clinical skills and trainee satisfaction.

**Conclusions**. Incorporating non-hypoxic apnea duration awareness into simulation-based airway management training significantly enhances both clinical skills and trainee satisfaction. This approach shows promise for improving critical aspects of airway management education.

## INTRODUCTION

Airway management stands as a fundamental skill in anesthesiology practice, with profound implications for patient safety and clinical outcomes (*Cook, Woodall & Frerk, 2011*). The complexity of this skill, combined with its critical nature, creates significant challenges in medical education, particularly in developing effective training methodologies that adequately prepare practitioners for real-world scenarios (*Green, Tariq & Green, 2016*). In recent years, simulation-based education has emerged as an invaluable tool in medical training, especially within the field of anesthesiology. This approach allows practitioners to develop and refine critical skills in a controlled environment without risking patient safety (*Gaba, 2004*). The effectiveness of simulation-based training has been particularly well-documented in procedural specialties, where repeated practice under standardized conditions can significantly improve technical proficiency (*McGaghie et al., 2011*).

A crucial yet often overlooked aspect of airway management is the concept of non-hypoxic apnea duration, defined as the time to reach an SpO2 threshold that is critical for avoiding significant hypoxemia during the apneic period. This physiological parameter provides essential context for clinical decision-making during airway management procedures (*Weingart & Levitan, 2012*). Conventional simulation approaches, however, frequently fail to incorporate this temporal element, potentially diminishing their clinical relevance and educational value (*Sun et al., 2017*).

Research has consistently demonstrated the positive impact of simulation-based training on airway management competency. Multiple studies have shown significant improvements in both technical skills and decision-making capabilities following structured simulation programs (*Kennedy et al., 2014*; *Lucisano & Talbot, 2012*). Meta-analyses of simulation-based medical education have further reinforced these findings, highlighting substantial enhancements in procedural competency across various medical specialties (*Cook et al., 2013*). Nevertheless, the integration of physiological parameters, particularly non-hypoxic apnea duration, remains notably understudied in the context of simulation-based airway management training (*Baker, Feinleib & O'Sullivan, 2016*).

While existing literature supports simulation training, there is a lack of research on incorporating time constraints in airway management training. This study addresses the limitation of non-hypoxic apnoea duration in airway management training. We sought to evaluate the impact of incorporating non-hypoxic apnoea duration into simulation scenarios, assess changes in clinical competence, and measure trainee satisfaction.

## METHODS

### Study design and setting

This quasi-experimental study was conducted at the Department of Anesthesiology, Shanghai General Hospital (Shanghai, China) between December 2022 and March 2023. The study was approved by the Ethics Committee of Shanghai General Hospital (No. 2022KY093) and registered in the Chinese Clinical Trial Registration Center (https://www.chictr.org.cn/) with the registration number ChiCTR2200065877 on 17/11/2022. Written informed consent was obtained from all participants prior to

enrollment. The study was conducted during scheduled simulation training sessions that were part of their curriculum. Each participant received a training session which lasted approximately 90 min, including a 30-minute pre-simulation briefing, a 30-minute simulation scenario, and a 30-minute debriefing session. Both the intervention and control groups received standardized training on airway management as part of their regular curriculum. This training included a 30-minute lecture on the principles of airway management, pre-oxygenation techniques, and the management of difficult airways. The lecture was followed by a hands-on workshop using the Laerdal Airway Management Trainer, where participants practiced the techniques under supervision.

## Participant selection

The study population comprised anesthesiology undergraduate students aged 18–25 years who had completed a preliminary two-month theoretical and bedside internship course. We excluded students who had less than three months of internship experience in the Department of Anesthesiology or previous experience in tracheal intubation. Previous experience was defined as having completed at least one prior tracheal intubation attempt.

Additionally, participants who withdrew during the study period or failed to complete the simulation within the specified timeframe were excluded from the final analysis.

## Randomization and study groups

Eligible participants were randomly assigned to either the non-hypoxic apnea duration application group or the conventional training group using a computer-generated random number sequence with a 1:1 allocation ratio. The allocation was implemented through sealed opaque envelopes to ensure concealment. An independent evaluator, who was blinded to group assignment, conducted all assessments throughout the study period.

## Simulation protocol

For the intervention group, an additional 15 min of focused training on non-hypoxic apnea duration awareness was incorporated into their pre-simulation briefing. This training was delivered in a lecture format, followed by a brief interactive session where participants could ask questions and receive clarification. The content of this additional training was standardized across all participants in the intervention group and focused on the concept of non-hypoxic apnea duration, its clinical significance, and how to manage patients within this time frame. The non-hypoxic apnea duration was defined as the time until $SpO_2$ reaches 90%, and participants were informed that this duration was 247 s in the simulated scenario (*Tang et al., 2022*) (Fig. 1). The control group received the standard training without the additional focus on non-hypoxic apnea duration. Instead, they participated in a 15-minute sham discussion on general anesthesia principles, which included a question-and-answer session to ensure consistency in the overall time commitment required from both groups. This structured approach ensured that the intervention was delivered consistently across all participants.

The simulation process began with each participant reviewing the patient's medical history and conducting a pre-anesthesia assessment, including physical examination of a standardized patient. Participants then developed their anesthesia plan and prepared the

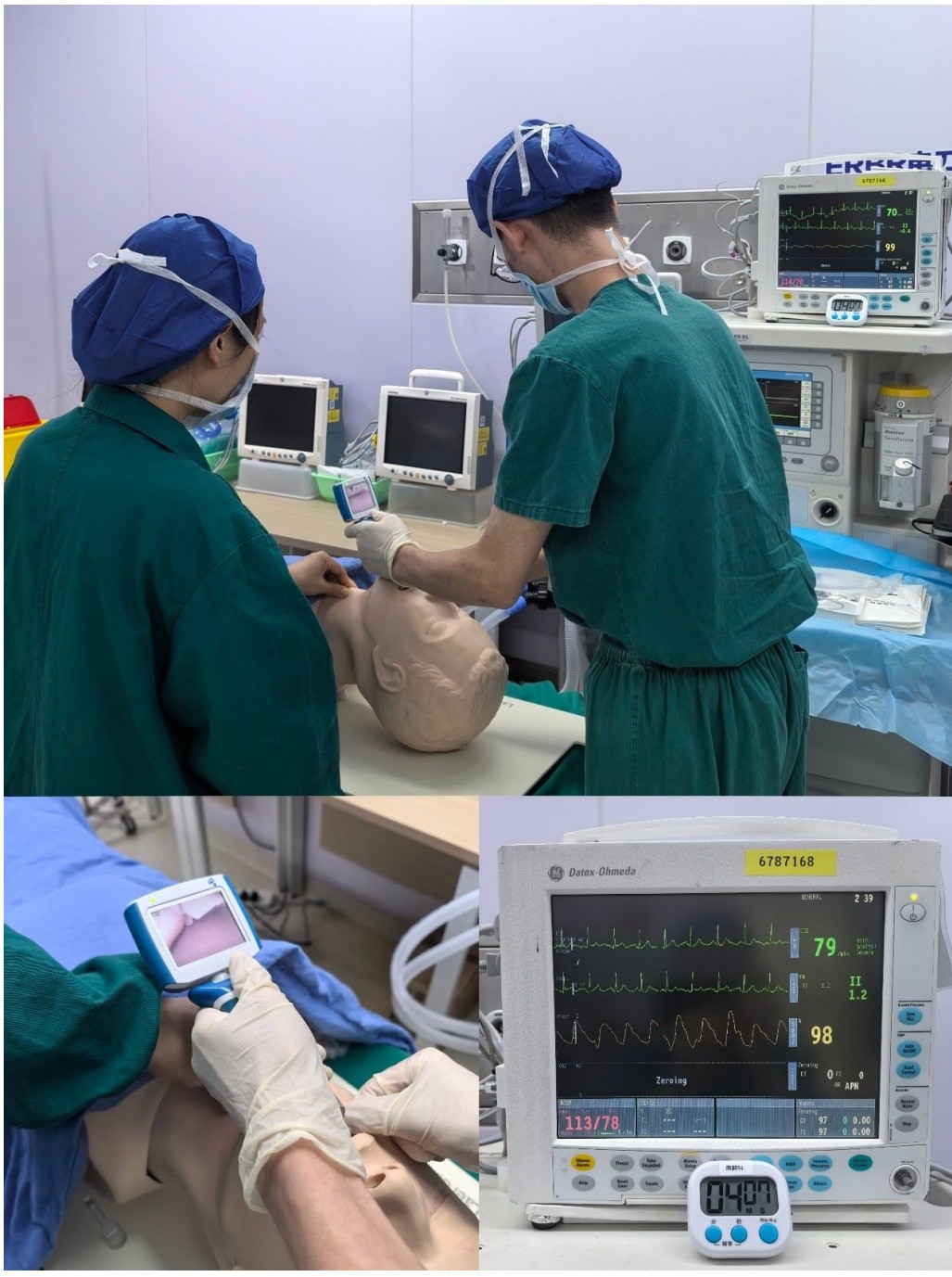

**Figure 1  Simulated difficult airway with the non-hypoxic apnea duration.** For the non-hypoxic apnea duration application group, the medical history included an additional parameter: the non-hypoxic apnea duration, defined as the duration until SpO2 reaches 90%, was set at 247 s and displayed as a real-time countdown during the simulation.

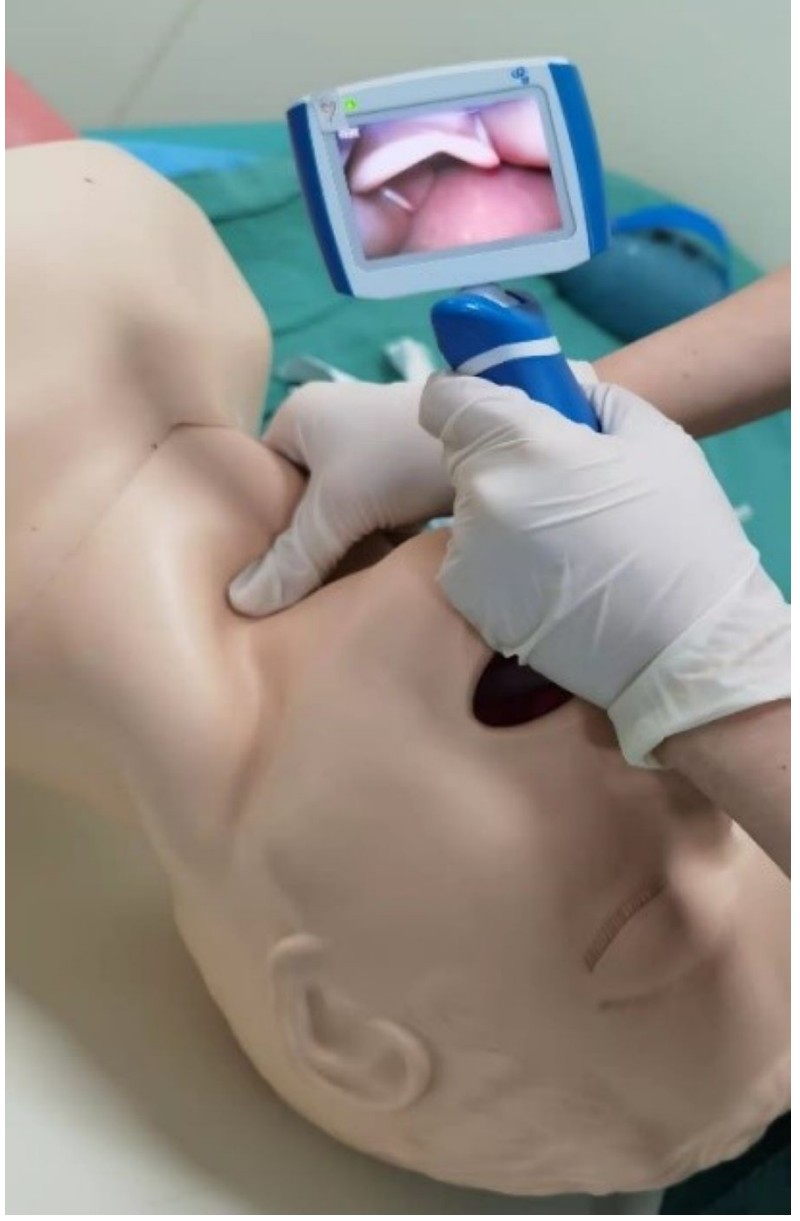

**Figure 2  Simulated difficult airway.** Before the trainee performed laryngoscopy, a simulated assistant was asked to simulate the difficult airway by applying the Sellick maneuver (increasing the force of pressure) to the airway of the simulated patient to block the visualization of the larynx.

necessary materials. During the simulation, a standardized difficult airway was created by having an assistant apply the Sellick maneuver with increasing pressure, thereby limiting glottic visualization during laryngoscopy (Fig. 2).

All simulations were conducted using a Laerdal Airway Management Trainer (Laerdal Medical Corporation, Gatesville, TX, USA). The simulation scenario presented an unexpected difficult airway in a patient under general anesthesia with a full stomach.

## Outcome assessment

The primary outcome measure utilized a modified Direct Observation of Procedural Skills (DOPS) assessment tool, which evaluated ten distinct aspects of performance: understanding of indications and relevant anatomy, pre-anesthesia visits and informed consent, pre-anesthesia preparation, aseptic procedure, pre-oxygenation technique, difficult airway management, help-seeking behavior, crisis management, team cooperation, and overall performance. Each aspect was scored on a scale of 1–9, where scores of 1–3 indicated below-expected performance, 4–6 indicated expected performance, and 7–9 indicated above-expected performance.

Secondary outcomes were assessed through an anonymous satisfaction questionnaire completed by participants following the simulation. The questionnaire evaluated ten domains of the training experience using a 5-point Likert scale (1 = strongly disagree to 5 = strongly agree). These domains encompassed learning interest, theoretical understanding, patient communication, airway management competency, procedural skills, crisis management, team cooperation, clinical practice relevance, simulation fidelity, and overall satisfaction (*Sparks et al., 2017*).

## Statistical analysis

Sample size calculation was performed with assumptions of an effect size of 0.8, α error of 0.05, and power of 0.8. Normally distributed continuous variables were expressed as mean ± standard deviation, and the independent sample $t$-test was implemented for intergroup comparisons. Non-normally distributed continuous variables were expressed as median (interquartile range), and non-parametric test was employed for intergroup comparisons. Categorical variables were expressed as number (percentage), and the chi-square test was used for intergroup comparisons. The statistical software SPSS 22.0 (IBM SPSS Statistics for Windows, IBM Corporation, Armonk, NY, USA) was utilized for the statistical analyses. $P < 0.05$ was considered to indicate a statistically significant difference.

## RESULTS

### Study population and baseline characteristics

A total of 31 trainees were initially assessed for eligibility. One trainee was excluded due to previous tracheal intubation experience. The remaining 30 trainees were randomly allocated into two equal groups ($n = 15$ each): the non-hypoxic apnea duration application group and the conventional training group (Fig. 3). All enrolled participants completed the study with no withdrawals or protocol deviations.

Baseline characteristics, including gender, age, and number of practices in general anesthesia and difficult airway management, showed no significant differences between the groups ($p > 0.05$) (Table 1).

### Modified DOPS scores

The non-hypoxic apnea duration application group demonstrated significantly superior performance in the modified DOPS assessment compared to the conventional training group (total score: 51.4 ± 4.4 *vs.* 43.0 ± 5.4, $P < 0.001$), as shown in Fig. 4. Domain-specific
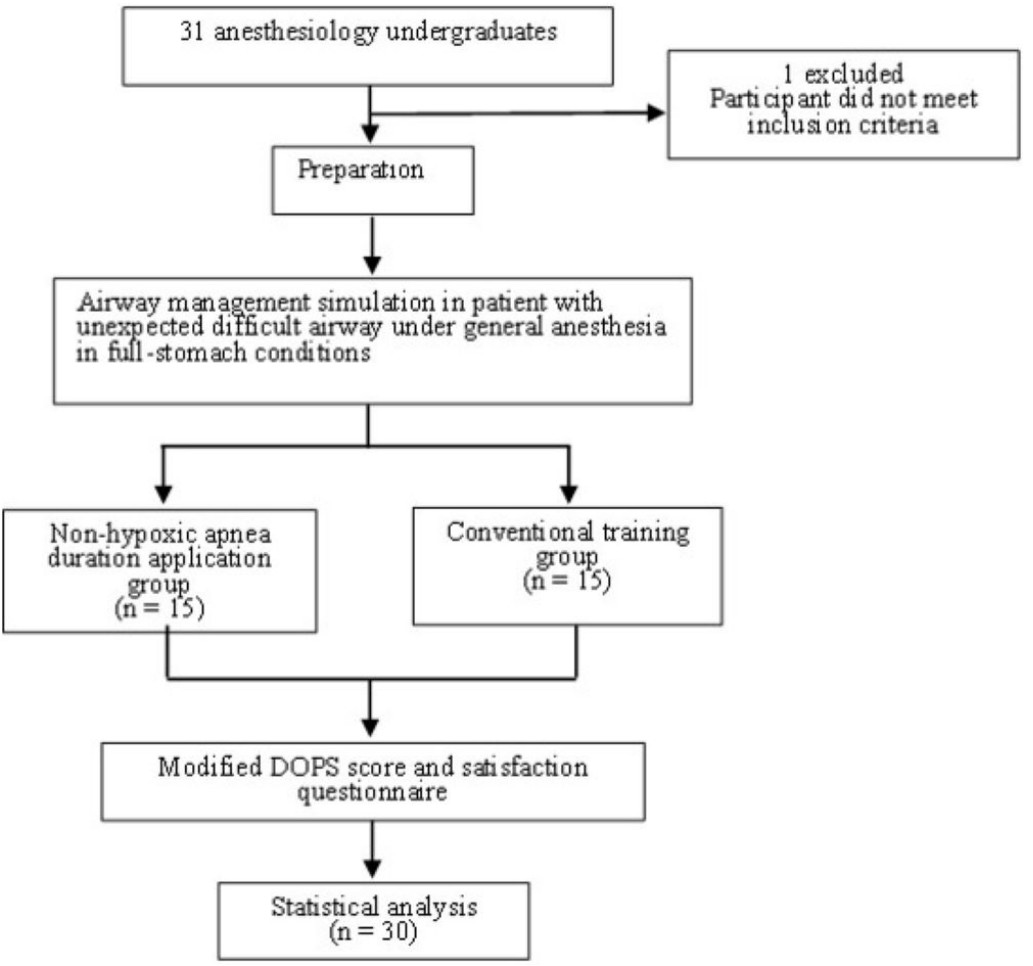

**Figure 3** Flow chart of the study.

analysis revealed significant improvements across multiple competencies. The non-hypoxic apnea duration application group demonstrated enhanced performance in pre-anesthesia preparation ($P = 0.028$), pre-oxygenation technique ($P = 0.003$), and difficult airway management ($P < 0.001$). Notably, substantial improvements were also observed in help-seeking behavior ($P < 0.001$) and crisis management ($P = 0.003$). The overall performance domain similarly showed significant advancement ($P = 0.003$), as illustrated in Table 2.

## Trainee satisfaction

The satisfaction questionnaire revealed significantly higher overall satisfaction in the non-hypoxic apnea duration application group (total score: $45.0 \pm 1.4$ *vs.* $43.1 \pm 2.0$, $P = 0.005$), as shown in Fig. 5. Detailed analysis of specific satisfaction components revealed particularly strong improvements in theoretical understanding and difficult airway management confidence (both $P < 0.001$). Clinical decision-making capabilities also showed significant enhancement ($P = 0.008$), as illustrated in Table 3.

**Table 1    Baseline characteristics.**

| Characteristic | Non-hypoxic apnea duration group (Mean ± SD) | Conventional training group (Mean ± SD) | *p*-value |
|---|---|---|---|
| Gender (M/F) | 8/7 | 9/6 | 0.563 |
| Age (years) | 22.3 ± 1.2 | 21.9 ± 1.4 | 0.422 |
| Number of general anesthesia practices | 10.4 ± 2.3 | 9.8 ± 2.0 | 0.215 |
| Number of difficult airway practices | 3.2 ± 1.5 | 3.0 ± 1.4 | 0.389 |
| Preliminary procedural test score | 82.1 ± 6.2 | 80.5 ± 5.9 | 0.152 |
| Preliminary theoretical test score | 78.4 ± 5.8 | 77.8 ± 6.1 | 0.189 |

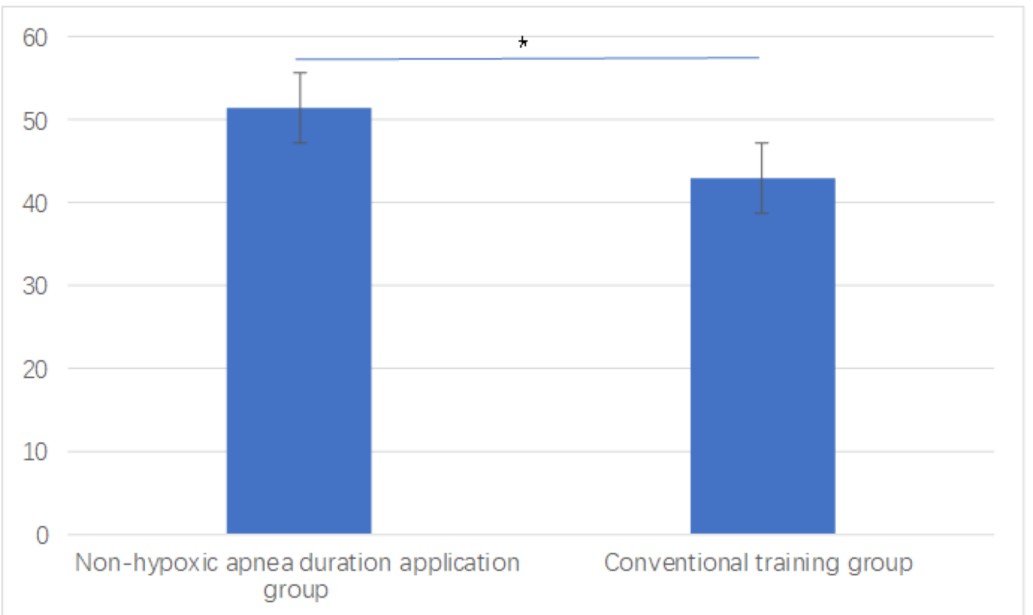

**Figure 4    Modified DOPS score between groups.** *$P < 0.05$ indicates statistical significance.

Both groups reported positive overall satisfaction with the airway management simulation training program, with consistently high scores across most domains of the satisfaction questionnaire.

## DISCUSSION

This study demonstrates the potential benefits of incorporating non-hypoxic apnea duration awareness into airway management simulation training. Presenting apnea duration in a quantifiable and objective manner led to significant improvements in technical performance and learner satisfaction compared to conventional training methods. These findings suggest that this approach can enhance the training outcomes and clinical skill development of anesthesiology trainees.

**Table 2  Modified DOPS scores.**

| Item | Non-hypoxic apnea duration application group (*n* = 15) | Conventional training group (*n* = 15) | *P*-value |
|---|---|---|---|
| Understanding of indications and relevant anatomy (9) | 5.0 (5.0, 5.5) | 5.0 (5.0, 6.0) | 0.108 |
| Pre-anesthesia visit and obtain informed consent (9) | 5.0 (5.0, 6.0) | 4.0 (3.0, 4.0) | <0.001[*] |
| Appropriate pre-anesthesia preparation (9) | 5.0 (5.0, 6.5) | 4.0 (4.0, 5.0) | 0.028[*] |
| Aseptic procedure (9) | 6.0 (5.0, 6.0) | 6.0 (5.0, 6.0) | 0.717 |
| Technical ability of pre-oxygenation (9) | 5.0 (5.0, 6.0) | 4.0 (3.5, 5.0) | 0.003[*] |
| Technical ability of managing difficult airway (9) | 5.0 (5.0, 6.0) | 4.0 (3.0, 4.0) | <0.001[*] |
| Seek help when appropriate (9) | 6.0 (6.0, 6.0) | 3.0 (3.0, 4.0) | <0.001[*] |
| Crisis management ability (9) | 4.0 (3.0, 4.0) | 3.0 (3.0, 4.0) | 0.566 |
| Team cooperation awareness (9) | 3.0 (3.0, 4.0) | 3.0 (3.0, 4.0) | 0.472 |
| Overall performance (9) | 6.0 (5.0, 6.0) | 5.0 (4.0, 5.0) | 0.003[*] |
| Total score (90) | 51.4 ± 4.4 | 43.0 ± 5.4 | <0.001[*] |

**Notes.**

The total score is expressed as mean ± standard deviation, whereas others are expressed as median (interquartile range).

[*]$P < 0.05$ indicates statistical significance.

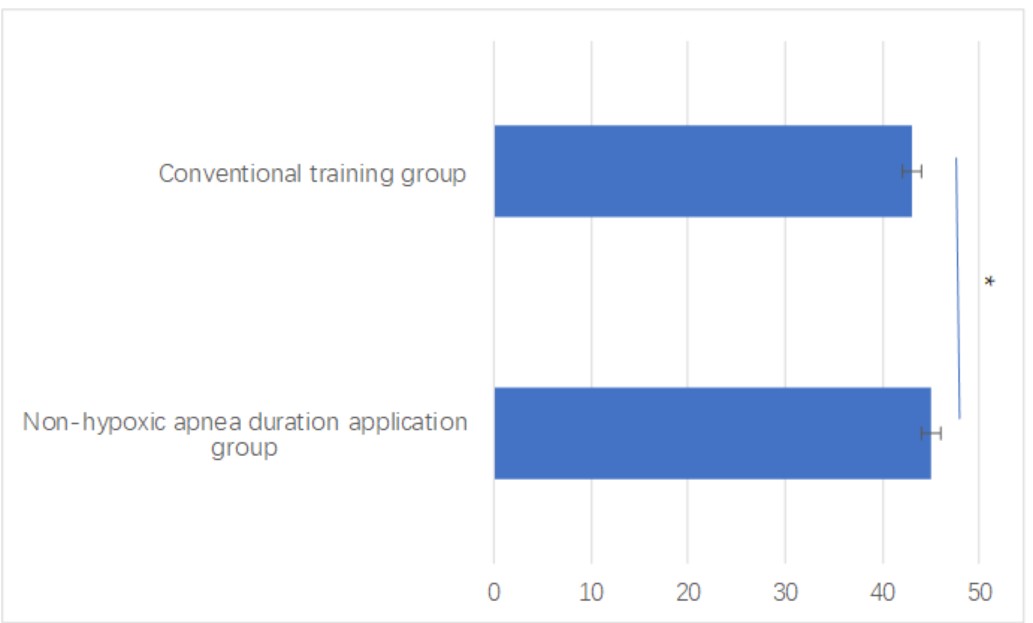

**Figure 5  Satisfaction questionnaire score between groups.** [*]$P < 0.05$ indicates statistical significance.

In the non-hypoxic apnea duration application group, the display of apnea duration appeared to motivate trainees to conduct more thorough pre-anesthetic evaluations, with particular attention to respiratory assessment and risk stratification. This enhanced focus on preoperative planning aligns with best practices in current anesthesiology education, where systematic risk assessment is increasingly emphasized (*Besnier et al., 2024*). Trainees

**Table 3  Satisfaction questionnaire scores.**

| Item | Non-hypoxic apnea duration application group ($n = 15$) | Conventional training group ($n = 15$) | $P$-value |
|---|---|---|---|
| Improvement of learning interests (5) | 5.0 (4.0, 5.0) | 5.0 (5.0, 5.0) | 0.417 |
| Benefits to understanding theories (5) | 5.0 (5.0, 5.0) | 4.0 (4.0, 4.5) | <0.001[*] |
| Improvement of communication skills with patients (5) | 3.0 (3.0, 4.0) | 4.0 (3.0, 4.0) | 0.720 |
| Improvement of ability of managing difficult airway (5) | 5.0 (5.0, 5.0) | 4.0 (4.0, 4.0) | <0.001[*] |
| Improvement of procedural skills (5) | 5.0 (4.0, 5.0) | 4.0 (4.0, 5.0) | 0.277 |
| Improvement of crisis management ability (5) | 5.0 (4.0, 5.0) | 4.0 (4.0, 5.0) | 0.281 |
| Improvement of team cooperation awareness (5) | 4.0 (4.0, 4.0) | 4.0 (3.0, 4.0) | 0.224 |
| Benefit to future clinical practice (5) | 5.0 (5.0, 5.0) | 5.0 (4.0, 5.0) | 0.203 |
| High-fidelity simulation (5) | 4.0 (4.0, 5.0) | 4.0 (4.0, 5.0) | 0.555 |
| Overall satisfaction (5) | 5.0 (5.0, 5.0) | 5.0 (5.0, 5.0) | 1.000 |
| Total score (50) | 45.0 ± 1.4 | 43.1 ± 2.0 | 0.005[*] |

**Notes.**

The total score is expressed as mean ± standard deviation, whereas the others are expressed as median (interquartile range).

*$P < 0.05$ represents statistical significance.

demonstrated improved pre-anesthetic preparation behaviors, including proactively arranging alternative airway management tools such as laryngeal masks, lighted stylets, and fiber-optic bronchoscopes. This comprehensive equipment preparation reflects a deeper understanding of the potential challenges associated with difficult airway management (*Karamchandani et al., 2021*).

A comprehensive grasp of apnea duration contributed to improved technical skills, particularly in preoxygenation techniques and positional optimization, highlighting how quantitative risk indicators can enhance clinical decision-making. Similar findings have been observed in other areas of medical simulation, where objective performance metrics have been shown to improve surgical proficiency (*Sheehan et al., 2019*). Furthermore, the intervention group exhibited a significantly higher incidence of appropriate help-seeking behaviors, suggesting that trainees developed better situational awareness and recognition of personal limitations, which are crucial skills in modern medical practice (*Su & Zeng, 2023*).

The development of anesthesiology education remains constrained by various factors, including reduced clinical exposure for students, heightened societal focus on patient safety, and an increasing demand for standardized assessment methods. Simulation-based training has thus become a key component in addressing these challenges (*Beutler Crawford, Johnson & Evans, 2023*). However, achieving high fidelity in simulation experiences remains a persistent challenge in medical education. Our findings suggest that incorporating quantitative clinical indicators, such as non-hypoxic apnea duration, can enhance the realism and educational value of simulation scenarios (*Cooper et al., 2012*). The integration of clinical metrics like apnea duration provides trainees with objective benchmarks that closely mirror real-life patient responses, offering critical insights beyond what conventional simulation alone can provide. Such metrics encourage trainees to engage in more rigorous

preparatory behaviors, recognize high-risk scenarios, and make data-driven adjustments during procedures. In doing so, quantitative indicators serve not only to increase the fidelity of simulation training but also to bridge the gap between theoretical knowledge and real-world clinical decision-making, ultimately fostering a higher level of clinical competence.

Selecting appropriate assessment tools is another critical consideration in medical education research. While workplace-based assessments have gained attention, their systematic validation remains an ongoing process (*Tanaka et al., 2024*). In this study, a modified DOPS was employed as an assessment tool, building upon existing evidence supporting its effectiveness in evaluating procedural skills (*Castanelli et al., 2019*). The significant improvements observed across multiple DOPS domains provide additional support for its practical application in anesthesiology education. This tool objectively documents trainee progress, offering a solid foundation for future research and enabling an evidence-based approach to tracking skill development over time.

There are several limitations in this study. First, the relatively small sample size and single-center design may restrict the generalizability of the findings. Smaller samples are more susceptible to variability and may not fully represent the diverse learning environments and trainee backgrounds found in larger, multi-institutional studies. Additionally, the short observation period limits the ability to assess the impact on long-term skill retention and clinical performance, which are crucial for understanding the sustained effectiveness of the training intervention. Moreover, although the modified DOPS provided a structured assessment framework, its application in undergraduate anesthesiology education lacks extensive validation. This limitation raises questions about the tool's reliability and generalizability in broader educational contexts. Future studies should address these limitations by conducting larger-scale, multi-center research, extending follow-up periods to better evaluate skill retention, and refining assessment methods to ensure accuracy and reproducibility. Despite these limitations, the findings hold substantial significance for anesthesiology education, highlighting potential pathways to enhance both the technical skills and clinical competencies of trainees.

## CONCLUSIONS

The incorporation of non-hypoxic apnea duration awareness into simulation-based airway management training significantly enhances both clinical skills and trainee satisfaction. This novel approach shows promise for improving critical aspects of anesthesiology education.

### Funding
The authors received no funding for this work.

### Competing Interests
The authors declare there are no competing interests.

## Author Contributions

- Lijun Tang conceived and designed the experiments, performed the experiments, prepared figures and/or tables, authored or reviewed drafts of the article, and approved the final draft.
- Yi Zhang conceived and designed the experiments, performed the experiments, analyzed the data, prepared figures and/or tables, and approved the final draft.
- Lianhua Chen conceived and designed the experiments, analyzed the data, prepared figures and/or tables, authored or reviewed drafts of the article, and approved the final draft.
- Jinbao Li performed the experiments, analyzed the data, authored or reviewed drafts of the article, and approved the final draft.
- Shiwei Huang analyzed the data, prepared figures and/or tables, and approved the final draft.
- Bin Fang performed the experiments, analyzed the data, prepared figures and/or tables, authored or reviewed drafts of the article, and approved the final draft.
- Xiaojing Huang conceived and designed the experiments, analyzed the data, prepared figures and/or tables, authored or reviewed drafts of the article, and approved the final draft.
- Lina Huang performed the experiments, authored or reviewed drafts of the article, and approved the final draft.

## Human Ethics

The following information was supplied relating to ethical approvals (i.e., approving body and any reference numbers):

Shanghai General Hospital, Shanghai Jiao Tong University School of Medicine granted Ethical approval to conduct the study within its facilities (Ethical Application Ref: No. 2022KY093).

## Data Availability

The raw measurements are available in the Supplementary Files.

## Supplemental Information

Supplemental information for this article can be found online at http://dx.doi.org/10.7717/peerj.19555#supplemental-information.

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
