# Peer review of "Advancing anesthesiology trainee proficiency in airway management via simulation-based training: a non-hypoxic apnea duration approach"

_PeerJ, doi:10.7717/peerj.19555_

## Round 0.1 · original submission · Major Revisions

Address all the comments of the reviewers, in particular the commens from Reviewer 1

Reviewer 1 ·

Basic reporting

Thank you for the opportunity to review the manuscript by Tang et al. They describe an educational setting where anesthesiology undergraduates in Shanghai were randomized during their airway management simulation to either receive the default experience or have additional training on non-hypoxic apnea duration. The authors should be commended on undertaking such a study and I would enjoy seeing more in the future. This appears to be a very relevant manuscript for the training of anesthesiology providers and students of various backgrounds who are involved with airway management. It appears to have a very robust design with convenience sampling. The findings are significant and may help guide future simulation trainings in our specialty. The largest weakness is the simulation and, furthermore, the intervention are mentioned so vaguely that in the current format, the manuscript provides no real benefit. For example, as an associate professor at my institution, when I train the next class of incoming residents to intubate in the simulation center, I am not certain if we have the same setup. I cannot be certain that we are covering similar material or how similar our settings are. Additionally, I don't know specifically what the authors did to teach about non-hypoxic apnea duration and how it was tested during simulation. Were handouts involved, was it in a lecture format, or was it more like a guided airway lab. Were sounds used to denote the falling saturations. Some other modality or process may be involved which isn't compatible with how we introduce our learners to airway manipulation. More description is necessary in the methods section to describe the training and the simulation. Supplemental content may be necessary depending on the complexity. A manuscript that I have submitted for publication included YouTube URLs for videos we uploaded to help readers visualize our intervention if they were so inclined.

Experimental design

Unable to know for certain.

Validity of the findings

Looks fine.

Additional comments

Potential references that should be added:
1. Blaine et al. Training Anesthesiology Residents to Care for the Traumatically Injured in the United States. Anesth Analg. 2023 May 1;136(5):861-876. doi: 10.1213/ANE.0000000000006417.
2. Komasawa et al. Simulation-based Airway Management Training for Anesthesiologists – A Brief Review of its Essential Role in Skills Training for Clinical Competency. J Educ Perioper Med. 2017 Oct 1;19(4):E612.
3. Maurya et al. Simulation in airway management teaching and training. Indian J Anaesth. 2024 Jan 18;68(1):52–57. doi: 10.4103/ija.ija_1234_23
4.
Abstract
Results: This is too brief. I know there are word limits, but this is so brief, it is hard to know what exactly the outcome was if they were perusing abstracts to see if the article was interesting or relevant to their teaching.
Introduction
Line 85: Is this oxygen saturation standardized yet? Kim et al used 95% in their article and Firdous et al used 93%.
Kim et al. Effect of positive end expiratory pressure on non-hypoxic apnea time and atelectasis during induction of anesthesia in infant: A randomized controlled trial. Paediatr Anaesth. 2024 Nov;34(11):1146-1153. doi: 10.1111/pan.14965. Epub 2024 Jul 9. PMID: 38980197.
Firdous et al. Compare the Mean Non Hypoxic Apnea Duration of 20° Head up with Conventional Supine Position during Pre-oxygenation in Patients Undergoing Elective Surgery. Pakistan Journal of Medical & Health Sciences, 16(07), 397. https://doi.org/10.53350/pjmhs22167397
Methods
Lines 108-112: Some description of how participants were selected/enrolled should be included here. It seems like based on the intervention that this was a convenience sampling. Were these just students who were available during rounds or was this part of some sort of formal didactic time. The description should help the reader understand what sort of time commitment would be required to see similar results if they applied the intervention at their institution.
Lines 108-117: Somewhere in here, there needs to be a description on what training participants had received. Since it sounds like only one iteration of the simulation occurred in the study (unclear based on writing), there must have been some additional training provided to the intervention group. Was this in lecture format or some sort of workshop? How long did the intervention group receive training? Were they allowed to ask questions? Was the training standardized? Did the control arm receive some sort of sham instruction or place-holder discussion where they could ask questions?
Line 110: How was previous experience defined? Was it any single previous attempt or something more?
Lines 119-130: What equipment was available? How long was the simulation duration or did it depend on something? How were vitals displayed? Was oxygen saturation the only vital displayed and was it only displayed in the intervention group? How was the end of the simulation determined? Was the only difference between groups the oxygenation display? Did it also have a sound where the pitch varied with oxygen saturation like in the operating room?
Discussion
Lines 181-181: This opening line to the discussion section, as written, is false.
Line 183: Traditional is probably not the correct adjective here since Miller made his laryngoscope in 1941, Machintosh in 1943, and the first airway simulator wasn’t invented until the 1960’s. Also, traditional at the author’s institution may not be traditional at other institutions. Maurya et al might also disagree with the authors’ use of the word and would also be another great reference for the manuscript.
Maurya et al. Simulation in airway management teaching and training. Indian J Anaesth. 2024 Jan 18;68(1):52–57. doi: 10.4103/ija.ija_1234_23
Lines 207-208: Without more information on the studied intervention, it is hard to know if the author’s statement is true. If only a lecture was given on non-hypoxic apnea duration, than it would seem that incorporation in this simulation was useful as an assessment tool for skills already acquired, and that the study showed that a lecture was sufficient to “enhance the realism and educational value.”
Lines 180-235: Some of the discussion should refer more to the literature about what kind of skill set was demonstrated in this study and best methods for acquiring. Recommend Blaine et al. if the authors don’t have another source. Other scales such as the T-NOTECHS score may be equally as helpful in this scenario
Blaine et al. Training Anesthesiology Residents to Care for the Traumatically Injured in the United States. Anesth Analg. 2023 May 1;136(5):861-876. doi: 10.1213/ANE.0000000000006417.

Table 1: These characteristics need more explanation in the text. Especially since prior experience was supposed to be an exclusion criteria.
Table 2: Some explanation will need to be given on why the authors’ thought that the preanesthesia visit and obtaining informing consent was affected by the intervention.
Figure 2: Does this mean the non-hypoxic apnea indicator was just a timer?

Annotated reviews are not available for download in order to protect the identity of reviewers who chose to remain anonymous.

Reviewer 2 ·

Basic reporting

This quasi-experimental study evaluates the impact of integrating non-hypoxic apnea duration awareness into simulation-based airway management training for anesthesiology trainees. It effectively shows the potential benefits of incorporating physiological parameters into education. The superior performance of the intervention group underscores the value of integrating physiological awareness elements into the training content.

Below, you can find some of my evaluations and recommendations for the study.

Introduction

Please provide more details about non-hypoxic apnea duration awareness training. Are there any similar studies conducted on this topic in the field of education? What is the gap in this area, and what are its underlying causes?

Experimental design

Methods

How did you determine the sample size, and how did you justify it? What was your reference values?
How did you ensure that the assistant performing the Sellick maneuver did not introduce bias between the groups?

Validity of the findings

Intervention is not very clear? Was training provided to the intervention group, or was an additional problem simply added to their scenarios?
Please provide more details about intervention.

·

Basic reporting

No comment

Experimental design

No comment

Validity of the findings

No comment

Additional comments

The study is relevant for teaching clinical skills in difficult airway management. The authors acknowledge the limitations of the study, which needs to have its sample expanded and applied in other scenarios. Promising results.

---

## Round 0.2 · Minor Revisions

Please address the remaining reviewer comments.

Reviewer 1 ·

Basic reporting

Thank you for the opportunity to re-review the manuscript by Tang et al. They have significantly improved the abstract, the technical writing, and the amount of detail. They have also faithfully addressed the reviewers’ questions/comments. I think that only very minor changes should be made for clarity, and further peer review is not necessary.

Reviewer #2 Comments – Second comment under experimental design
The authors state that the assistant was blinded to group assignment. How is this possible? Did they just have their back to the screen with the timer on it? I think the authors would be better off omitting the word “blinded” and just specifically stating what the assistant did not have knowledge of.
Abstract – Lines 67-69
I think this should be reworded since this was performed on simulators and not actual patients. It should state that it “may enhance” or that the study is “highly suggestive”. There is the possibility that the skills in simulation are not transferable to the clinical environment. In this specific case, it may be because of increased anxiety when caring for real patients or that the additional factors in clinical care cause the clinician to be distracted from the non-hypoxic apnea time.
Introduction – Line 88
Since 90% is mentioned in the methods section for this study, this should be written exactly as the authors worded in the response to the reviewers. Recommend “…non-hypoxic apnea duration, defined as the time to reach an SpO2 threshold that is critical for avoiding significant hypoxemia during the apneic period.”
Methods - Line 110
How many training sessions did each participant receive? If correct, recommend rewording to state, “Each participant received a training session which lasted approximately 90 minutes…”
- Simulation protocol
For chronological purposes, I would invert the order of the paragraphs. I would put the 3rd paragraph in front of the second, and the second in front of the first.

Experimental design

no comment

Validity of the findings

no comment

---

## Round 0.3 · accepted · Accept

Dear Authors, Thank you for your revised manuscript which has been re reviewed and accepted. Congratulations!

Reviewer 1 ·

Basic reporting

Perfect

Experimental design

Perfect

Validity of the findings

Perfect

Additional comments

Perfect